# Structural basis for peroxidase encapsulation inside the encapsulin from the Gram-negative pathogen *Klebsiella pneumoniae*

Jesse A. Jones [1,2], Michael P. Andreas[1,2] & Tobias W. Giessen [1] ✉

Encapsulins are self-assembling protein nanocompartments capable of selectively encapsulating dedicated cargo proteins, including enzymes involved in iron storage, sulfur metabolism, and stress resistance. They represent a unique compartmentalization strategy used by many pathogens to facilitate specialized metabolic capabilities. Encapsulation is mediated by specific cargo protein motifs known as targeting peptides (TPs), though the structural basis for encapsulation of the largest encapsulin cargo class, dye-decolorizing peroxidases (DyPs), is currently unknown. Here, we characterize a DyP-containing encapsulin from the enterobacterial pathogen *Klebsiella pneumoniae*. By combining cryo-electron microscopy with TP and TP-binding site mutagenesis, we elucidate the molecular basis for cargo encapsulation. TP binding is mediated by cooperative hydrophobic and ionic interactions as well as shape complementarity. Our results expand the molecular understanding of enzyme encapsulation inside protein nanocompartments and lay the foundation for rationally modulating encapsulin cargo loading for biomedical and biotechnological applications.

Intracellular compartmentalization enables cells to establish defined microenvironments for facilitating the precise regulation of metabolism in both time and space[1]. Effective spatial organization helps optimize crucial cellular functions, including the storage of nutrients and the sequestration of toxic or incompatible reactions and processes[1–3]. In prokaryotes, compartmentalization predominantly relies upon protein-based compartments and is realized at the individual enzyme or pathway level[1–5]. The two most prominent types of protein compartments found in prokaryotes are bacterial microcompartments (BMCs) and encapsulins[6–8]. While BMCs sequester short metabolic pathways and are involved in carbon fixation[9–11] and various catabolic processes[12–14], encapsulin function is more varied and usually relies on encapsulating a single type of cargo enzyme[7,15,16]. Encapsulins have been shown to be involved in iron storage[17,18], sulfur metabolism[19,20], and oxidative stress response[21,22].

Encapsulin shell proteins self-assemble into 18–42 nm protein shells consisting of 60, 180, or 240 identical subunits and exhibit icosahedral symmetry with triangulation numbers of T = 1 (T1), T = 3 (T3), or T = 4 (T4)[23]. It has been hypothesized that encapsulins may have originated from defective prophages whose capsid components were co-opted by the ancestral cellular host, an assertion supported by the fact that encapsulin shell proteins possess the HK97 phage-like fold found in a wide variety of viral capsid proteins[7,17,24,25]. Recent genome data mining efforts have resulted in the grouping of encapsulins into four separate families defined by sequence similarity, operon organization, and cargo encapsulation mechanism[7,26,27]. Family 1 is the most extensively characterized while the first studies focused on Family 2

---

[1]Department of Biological Chemistry, University of Michigan Medical School, Ann Arbor, MI, USA. [2]These authors contributed equally: Jesse A. Jones, Michael P. Andreas. ✉e-mail: tgiessen@umich.edu

encapsulins have only recently been reported[19,20,28]. Family 3 and Family 4 remain putative and currently lack experimental validation. Encapsulins are named for their ability to efficiently and selectively encapsulate dedicated cargo proteins[24]. Encapsulation is mediated by targeting motifs, found in all native cargo proteins, referred to as targeting domains (TDs) or targeting peptides (TPs)[15]. Removing TDs or TPs completely abolishes cargo loading[15]. Cargo encapsulation in all Family 1 systems is based on short TPs usually found at the cargo protein C-terminus[7,24,29]. Each subunit of the encapsulin shell contains one TP-binding site, however, based on cargo size and oligomerization state, experimentally observed cargo loading and TP occupancy are far below the theoretical maximum of 60 (T1), 180 (T3), or 240 (T4)[7,15]. TPs are usually connected to the cargo protein by a flexible linker with high glycine and proline content which likely minimizes steric clashes between adjacent cargo proteins within the shell. So far, only TP-shell interactions of ferroxidase cargo proteins (ferritin-like proteins and iron-mineralizing encapsulin-associated firmicute cargos) have been investigated structurally[17,18,24,30]. It was found that TP binding sites always reside on the luminal surface of encapsulin protomers with between 7 and 12 TP residues tightly interacting with a narrow binding pocket. A binding mode primarily based on hydrophobic interactions has been proposed where two conserved hydrophobic side chains – generally Leu, Ile, or Val – interact with hydrophobic patches within the TP-binding site. However, no detailed experimental analysis of the contributions of individual TP or TP-binding site residues towards cargo encapsulation has been reported. As TPs are highly modular and can be used as simple protein tags, encapsulin systems have received substantial interest from the synthetic biology and bioengineering communities as platforms for the rational development of biocatalytic nanoreactors, polyvalent vaccines, and targeted delivery modalities[31–37].

The most prevalent cargo type in encapsulin systems are dye-decolorizing peroxidases (DyPs) – all found in Family 1 operons[38]. DyPs are ubiquitous, often homohexameric, heme-containing enzymes, named for their ability to oxidize and degrade a wide range of synthetic dyes – with concomitant reduction of hydrogen peroxide to water[21,22,26]. DyP-containing encapsulin systems are widespread across bacterial phyla, including many prominent Gram-positive and Gram-negative pathogens[22,38–40]. Recent reports have highlighted their importance in the oxidative stress response of the Gram-positive pathogen *Mycobacterium tuberculosis* during infection[39]. No DyP encapsulin system from a Gram-negative pathogen has been studied to date. While DyP and encapsulin shell structures have recently been reported[21], the structural basis for DyP encapsulation inside encapsulin shells has so far remained elusive, primarily due to low DyP cargo loading observed for native DyP encapsulins, preventing visualization of the TP-shell interaction. DyPs have been proposed as useful promiscuous enzyme catalysts for various industrial applications – ranging from wastewater treatment to lignin degradation[41] – however, their native substrates and precise biological functions are currently unknown.

In this study, we report the structural and functional analysis of a DyP encapsulin system from a Gram-negative pathogen – *Klebsiella pneumoniae* (Kp). Using cryo-electron microscopy (cryo-EM), we determine the structure of the DyP-loaded Kp encapsulin shell (KpEnc) at 2.5 Å and the TP-shell interaction as well as the free DyP hexamer at 2.4 Å. By employing a non-native cargo-loading strategy aimed at maximizing TP occupancy, we resolve the TP-shell interaction in the Kp DyP encapsulin system at 2.4 Å. We provide direct visualization of the structural basis for cargo encapsulation in DyP encapsulins. Finally, we carry out a systematic and exhaustive mutagenesis screen to elucidate the contributions of individual TP and TP-binding site residues toward cargo encapsulation. In sum, our data reveal the molecular details and logic of DyP cargo loading inside encapsulin shells and provide a solid foundation for future

attempts at rationally engineering cargo loading for biomedical and biotechnological applications.

## Results

### Phylogenetic and computational analysis of enterobacterial DyP encapsulin operons and heterologous production of peroxidase-loaded KpEnc

Here, we focus on a DyP encapsulin system from the Gram-negative human pathogen *K. pneumoniae* (UniParc ID: UPI001261BC85). The genus *Klebsiella* is part of the *Enterobacteriaceae* family which contains a number of other notable pathogenic members, including *Escherichia*, *Salmonella*, *Shigella*, and *Enterobacter* species. Highly conserved DyP encapsulin operons can be found in all abovementioned enterobacterial genera (Fig. 1a and Supplementary Data 1 and Supplementary Data 2)[26]. In almost all cases, these operons are found in highly mobile regions of the genome flanked by various types of transposase genes and mobile genetic elements. In addition to the conserved two-gene operon encoding the DyP cargo and the encapsulin shell protein, a well-conserved gene annotated as a formate dehydrogenase (FDH) is found immediately upstream of the DyP gene (Fig. 1b). No TP is found in the FDH indicating that it does not represent an encapsulin cargo. FDHs are diverse and ubiquitous enzymes found in both prokaryotes and eukaryotes capable of utilizing a variety of electron acceptors, such as quinones and nicotinamides, to catalyze the reversible oxidation of formate to carbon dioxide[42–44]. As such, FDHs are generally involved in the catabolism of C1 compounds as well as various aspects of anaerobic metabolism[43]. Recently, FDHs have also been implicated in prokaryotic stress response[45]. Further, a well-conserved RNA polymerase sigma factor gene (RpoH-like) can be found downstream of the encapsulin gene. RpoH is a well-known regulator of the *E. coli* heat-shock response[46]. The conserved genome neighborhood surrounding the two-gene DyP encapsulin operon, points towards a function in stress response or detoxification.

Our bioinformatic analysis identified 1193 unique DyP cargo enzymes found in encapsulin operons. Aligning the 20 C-terminal residues of all identified DyPs revealed a strongly conserved GSLxIGSLK motif – representing the TP – only found in DyPs associated with an encapsulin (Fig. 1c and Supplementary Data 3). The *K. pneumoniae* DyP (KpDyP; UniParc ID: UPI0018699629) contains a C-terminal TP – GSLNIGSLK – very similar to this consensus motif.

Heterologous expression of the Kp two-gene core operon (KpDyP_Enc) in *Escherichia coli* BL21 (DE3) followed by protein purification via polyethylene glycol precipitation, size exclusion (SEC) and ion exchange chromatography, yielded readily assembled and DyP-loaded encapsulin shells. Analytical SEC and SDS-PAGE analysis showed clear co-elution of KpDyP (38.6 kDa) – easily detected by its 410 nm heme absorption – with KpEnc (28.8 kDa) at an elution volume characteristic for T1 encapsulin shells (Fig. 1d, e). Based on gel densitometry analysis, on average 12 copies of KpDyP are present per 60mer KpEnc shell, suggestive of encapsulation of two KpDyP hexamers per KpEnc. Subsequent negative stain transmission electron microscopy (TEM) analysis showed homogeneous encapsulin shells with a diameter of ca. 24 nm – in good agreement with previously reported T1 encapsulins – with visible internalized cargo (Fig. 1f and Supplementary Fig. 1). Dynamic light scattering (DLS) analysis further confirmed the size and integrity of purified KpDyP_Enc, yielding a Z-average diameter of 24.8 nm (Supplementary Fig. 2).

### Single particle cryo-EM analysis of the DyP-loaded encapsulin KpDyP_Enc

To investigate the structure and cargo-loading mechanism of the DyP-loaded encapsulin, single particle cryo-EM analysis was carried out on purified KpDyP_Enc (Supplementary Fig. 3 and Supplementary Table 1). The encapsulin shell was determined to 2.5 Å via icosahedral (I) refinement. In line with our previous analysis, the protein shell was

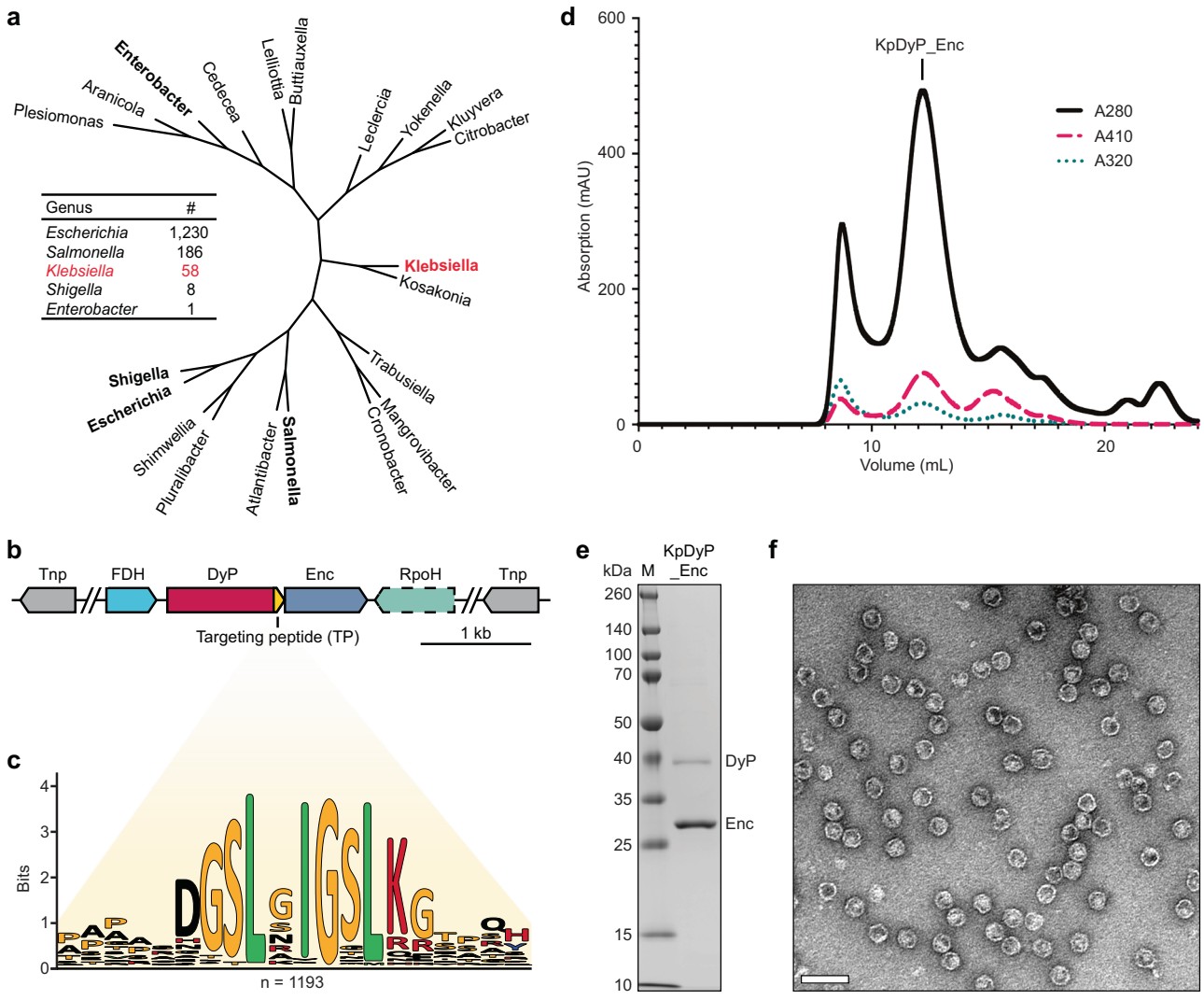

**Fig. 1 | Phylogenetic and bioinformatic analysis of enterobacterial DyP encapsulins and heterologous production of KpDyP_Enc. a** Representative phylogenetic tree of DyP encapsulins found in the *Enterobacteriaceae* family. Genera with prominent pathogenic members are shown in bold. The genus of interest for this study (*Klebsiella*) is shown in red. The inset highlights the number of distinct DyP encapsulins identified in the listed enterobacterial genera. **b** Representative *K. pneumoniae* DyP encapsulin operon. The generally four conserved genes of the operon are flanked by various types of transposases (Tnp, gray). FDH NAD-dependent formate dehydrogenase (light blue), DyP dye-decolorizing peroxidase cargo (red; targeting peptide (TP): orange), Enc Family 1 encapsulin shell protein (blue), RpoH RNA polymerase sigma factor (turquoise). Dashed outlines indicated mostly but not strictly conserved genes. kb kilobase. **c** Sequence logo for TPs found in DyP cargos highlighting the conserved targeting motif. n number of cargo sequences used. **d** Size exclusion chromatogram of KpDyP_Enc (Superose 6). The elution volume (12 mL) of the labeled main peak is consistent with a T1 encapsulin while the co-eluting 410 nm heme signal suggests DyP encapsulation. **e** SDS-PAGE analysis of purified KpDyP_Enc. This purification was repeated independently six times. Source data are provided as a Source Data file. **f** Negative stain transmission electron micrograph of purified KpDyP_Enc. This experiment was repeated independently five times. Scale bar: 50 nm.

found to be 24 nm in diameter and to consist of 60 identical subunits showing T1 icosahedral symmetry (Fig. 2a). The asymmetric unit contains one individual protomer that exhibits the HK97 phage-like fold consisting of the canonical axial domain (A-domain), peripheral domain (P-Domain), and extended loop (E-loop) (Fig. 2b)[25,47]. The shell contains two open pores at the five- and three-fold axes of symmetry, with a diameter of 9 Å and 3 Å, respectively (Fig. 2c–e and Supplementary Figs. 4 and 5)[48,49]. No open pore was observed at the two-fold axis of symmetry. Due to its size, the large five-fold pore represents the likely access point for small molecule DyP substrates to enter the interior of the compartment, as suggested previously[21]. The exterior of the five-fold pore is significantly negatively charged, the midpoint is relatively neutral, and the interior of the pore is positively charged

(Supplementary Fig. 5). The observed pore sizes and charges are in principle compatible with the transmission of a wide range of small molecules and do not meaningfully narrow down the range of potential native DyP substrates. This is particularly true when considering the fact that some Family 1 encapsulins have recently been reported to possess dynamic pores able to open and close based on environmental conditions and potentially other so far unknown stimuli[30,50].

Even though icosahedral (I) refinement yielded high-quality cryo-EM density for the encapsulin shell, DyP cargo could not be visualized because of compositional heterogeneity and conformational flexibility. This is almost always the case for cargo-loaded encapsulins due to the fact that cargo proteins – including DyPs – are generally flexibly tethered to the shell interior via TPs which allows for substantial cargo

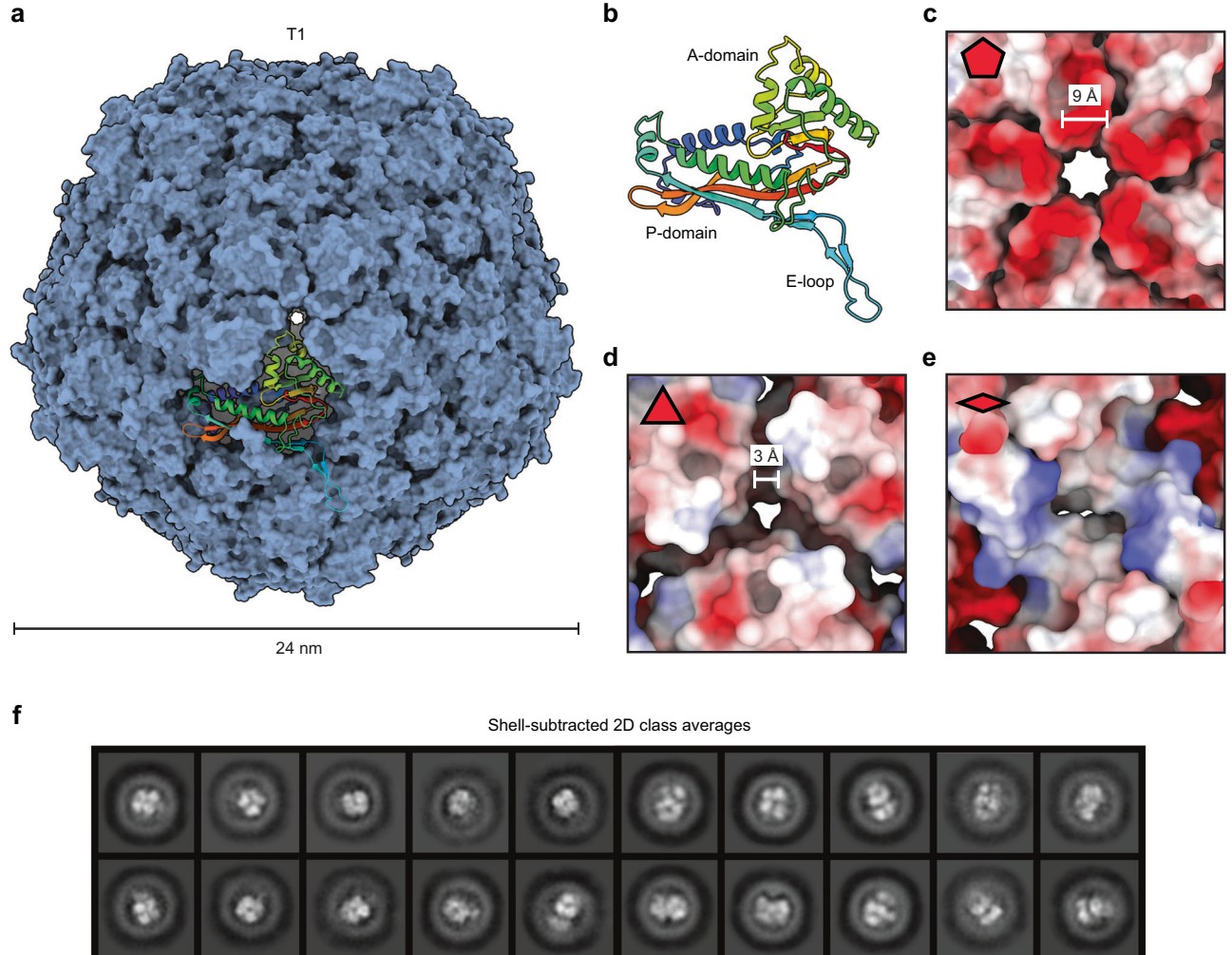

**Fig. 2 | Structure of the encapsulin shell. a** Surface representation of the KpEnc encapsulin shell – obtained from a KpDyP_Enc sample – viewed down the 5-fold symmetry axis with a single subunit shown in ribbon representation using rainbow coloring. **b** Ribbon diagram of the KpEnc protomer. **c** Electrostatic surface representation of the exterior view of the 5-fold pore with the pore diameter shown. **d** Electrostatic surface representation of the exterior view of the 3-fold pore with the pore diameter shown. **e** Electrostatic surface representation of the exterior view along the 2-fold pore. **f** Shell-subtracted 2D class averages of KpDyP_Enc highlighting distinct internalized DyP cargo assemblies.

mobility within the encapsulin compartment, thus preventing direct cargo structure determination[7]. However, an alternative approach, relying on computationally subtracting shell density, followed by 2D classification focused on the compartment interior, resulted in 2D classes with clear internalized densities (Fig. 2f). These densities are similar in size and shape to previously described DyP hexamers[39,51] with generally one or two putative hexamers present per shell. Moreover, due to the large size of DyP complexes and the concomitantly small number of DyP copies per shell, TP occupancy was low. Consequently, little TP signal could be observed in our cryo-EM map which prevented detailed visualization of the TP-shell interaction underlying DyP cargo loading. TP density in this dataset could not be improved using various computational data processing approaches, including symmetry expansion, 3D classification, and local refinements.

## KpDyP forms a catalytically active hexamer

To investigate the influence of encapsulation on KpDyP and to compare the activities of encapsulated and free enzyme, a C-terminally His-tagged KpDyP construct (40.6 kDa) was heterologously expressed and purified. Subsequent SEC analysis showed two main elution peaks at 16 mL – consistent with a KpDyP hexamer (ca. 244 kDa) – and 18 mL, likely representing lower-order KpDyP multimers or

monomers (Fig. 3a). This is in agreement with previous studies that reported encapsulin-associated DyPs[39,51], and DyPs in general[41], to exist as hexamers in solution. We did not observe any elution peaks consistent with dimers of hexamers as reported previously[21]. The characteristic 410 nm heme absorption was found to primarily co-elute with the hexamer peak, potentially indicating that hexamer formation encourages stable heme loading of KpDyP (Fig. 3a). As heme is necessary for catalytic activity, the KpDyP hexamer likely represents the primary catalytically competent state of the enzyme. Subsequent negative stain TEM analysis of the 16 mL fraction showed ca. 10 nm KpDyP complexes consistent with the expected size of a hexamer (Fig. 3b). No higher-order multimers could be observed in the 18 mL fraction (Fig. 3c). DLS analysis of the 16 mL fraction yielded a Z-average diameter of 13 nm, again suggesting a hexameric KpDyP assembly (Supplementary Fig. 6). Analytical SEC coupled with native PAGE analysis further confirmed the presence of multiple multimeric states including hexamers, dimers, and monomers (Supplementary Fig. 7).

Comparison of the A410 to A280 ratios of free KpDyP and encapsulated DyP (KpDyP_Enc) indicated a higher heme-to-protein ratio for free KpDyP (Fig. 3a and Supplementary Fig. 7). Subsequent heme content analysis yielded heme-loading values of 99% for KpDyP

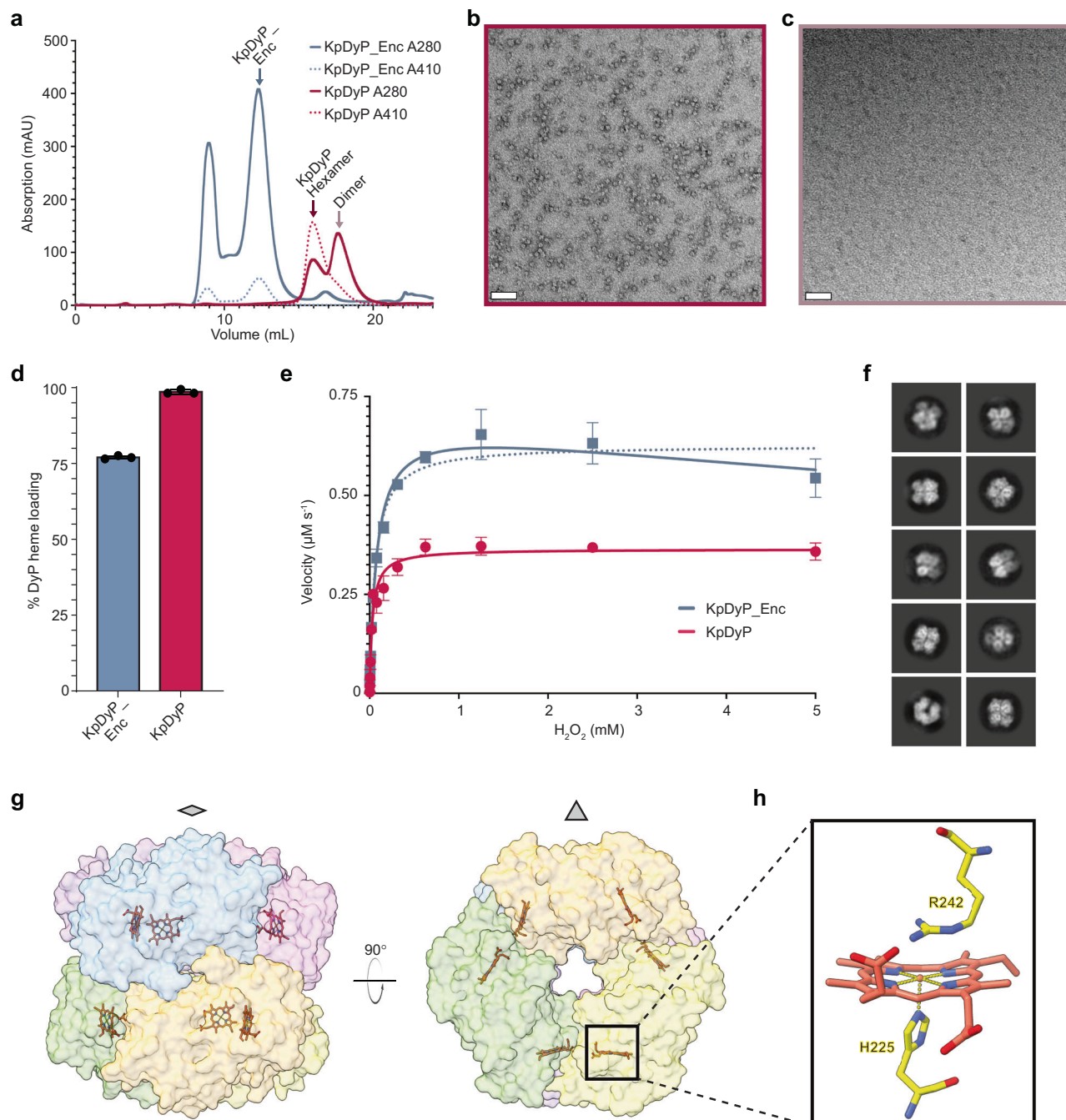

**Fig. 3 | Characterization of free KpDyP. a** Size exclusion chromatogram of free KpDyP (solid red line) compared with KpDyP_Enc (solid blue line) (Superdex 200). The respective main peaks are labeled. Dashed lines indicate heme absorption at 410 nm. **b** Negative stain TEM micrograph of free KpDyP (16 mL hexamer peak) highlighting homogeneous ca. 10 nm protein complexes. This experiment was repeated independently six times. **c** Negative stain TEM micrograph of the 18 mL free KpDyP peak (likely dimer). Scale bars: 50 nm. This experiment was repeated independently five times. **d** Heme loading of purified free KpDyP and encapsulated KpDyP_Enc determined via in vitro heme quantitation assay. Percentages refer to heme binding site occupancy, with one binding site per DyP subunit. Data are shown as mean values. Error bars represent standard deviation of three independent experiments. Source data are provided as a Source Data file. **e** Saturation

kinetics of encapsulated KpDyP_Enc (blue squares) and free KpDyP (red circles). Hydrogen peroxide ($H_2O_2$) was varied while ABTS was kept constant. Data are shown as mean values. Error bars represent standard deviation of three independent experiments. For KpDyP_Enc, a standard Michaelis-Menten fit is shown as a dashed line while a competitive substrate inhibition fit is shown as a solid line. Source data are provided as a Source Data file. **f** Representative 2D class averages of free KpDyP. **g** Model of the KpDyP hexamer viewed down the two-fold (left) and three-fold (right) axes of symmetry as determined via cryo-EM. Individual DyP subunits are shown in transparent surface representation with distinct colors. Heme groups are shown as stick models (tomato red). **h** Heme group of a DyP subunit shown in stick representation. The proximal histidine (H225) and distal arginine (R242) are shown.

and 75% for KpDyP_Enc under heterologous expression conditions (Fig. 3d). To confirm and compare the peroxidase activities of KpDyP and KpDyP_Enc, standard saturation kinetics analysis using a fixed amount of the known synthetic DyP substrate 2,2′-Azino-bis(3-

ethylbenzothiazoline-6-sulfonic acid) (ABTS) and varying amounts of hydrogen peroxide were carried out[21,51]. Enzyme concentrations were standardized based on heme content to ensure equivalent amounts of active enzyme. Free KpDyP and encapsulated KpDyP_Enc were found

to exhibit Michaelis-Menten constants ($K_m$) of 30 and 61 µM, and catalytic turnover numbers ($k_{cat}$) of 0.61 and 1.0 s$^{-1}$, respectively. At higher peroxide concentrations, KpDyP_Enc exhibited moderate substrate inhibition. The observed higher $K_m$ for KpDyP_Enc is similar to other reported cases of encapsulated enzymes which often exhibit increased $K_m$ values due to the protein shell acting as a diffusion barrier for substrates[17,19,20,22]. Because the native substrates of DyPs are not known, dyes like ABTS have often been used to confirm their catalytic activity[40], resulting in a fairly wide range of reported kinetic parameters[41]. The $k_{cat}$ and $K_m$ values of free and encapsulated KpDyP determined in this study fall within the range of values found in the literature.

## Single particle cryo-EM analysis of the KpDyP hexamer

To gain a deeper understanding of the molecular structure of KpDyP, single particle cryo-EM was carried out. The structure of the free KpDyP hexamer was determined to 2.4 Å via D3 symmetry refinement followed by local refinement (Supplementary Fig. 8 and Supplementary Table 1). The overall structure corroborates our biophysical and imaging analyses and confirms that free KpDyP exists as a catalytically competent hexamer (Fig. 3f, g). Cryo-EM density for a non-covalently bound heme cofactor – one per subunit – is clearly visible (Supplementary Fig. 8). The heme group is coordinated by residue His225, representing the proximal ligand to the heme iron. Arg242 is the residue closest to the heme iron on the distal face of the heme group (Fig. 3h). This arrangement is in good agreement with other DyP structures and reflects a mostly conserved active site organization across various DyPs[41]. Moreover, the heme cofactor is substantially buried, with the entrance to the active site partially obstructed by an adjacent subunit at the two-fold interface (Supplementary Fig. 9). The KpDyP surface was found to be predominantly negatively charged (Supplementary Fig. 9), which is in agreement with its theoretical isoelectric point (pI) of 4.65[52]. Low pI values are often associated with increased stability under acidic conditions[53]. Many DyPs have in fact been shown to be optimally active at acidic pH – at least when assayed with non-native synthetic substrates[40,41,54]. The entrance to the active site is also highly negatively charged. In line with the predicted disordered state of the C-terminus, the last 60 residues – including the TP – are unresolved, which is generally observed for encapsulin cargo proteins in solution[17,18,55].

## Cryo-EM analysis of highly SUMO-loaded KpEnc reveals the structural basis for cargo encapsulation in DyP encapsulins

To overcome the problem of low DyP cargo loading mentioned above and determine the molecular basis for cargo encapsulation, specifically the TP-shell interaction, in DyP encapsulins, we set out to increase cargo loading and thus TP occupancy by utilizing a small non-native cargo genetically fused to the native KpDyP TP[56]. In particular, the C-terminal 22 residues of KpDyP – containing the putative TP – were fused to the C-terminus of the 11 kDa monomeric small ubiquitin-like modifier (SUMO) protein via an 11-residue flexible glycine-serine linker (Supplementary Fig. 10 and Supplementary Table 1). After heterologous expression and purification, single-particle cryo-EM analysis was carried out on SUMO-loaded KpEnc (SUMO-TP_Enc). The consensus icosahedral (I) refinement yielded a resolution of 2.4 Å and the resulting cryo-EM density showed substantially improved signal for the KpDyP TP (Fig. 4a, b and Supplementary Fig. 11). As such, a ten-residue TP could be confidently modeled (SGSLNIGSLK) highlighting the details of the TP-shell interaction (Fig. 4a–c). In particular, TP-shell binding is mediated by a combination of hydrophobic, ionic, and H-bonding interactions. The three hydrophobic residues Leu340, Ile342, and Leu345 of the KpDyP TP interact with a geometrically complementary hydrophobic pocket located on the interior surface of the encapsulin shell protein (Fig. 4c). Additionally, Lys346 forms a salt bridge with an aspartate side chain carboxyl (Asp38) which is part of

the shell protein P-domain (Supplementary Fig. 12a). Multiple hydrogen bonds also contribute to TP binding, including interactions between the TP backbone (Leu345 and Gly343) and encapsulin shell residues Arg34 and Asp229 (Supplementary Fig. 12b). Finally, an intramolecular hydrogen bond between the two TP residues Asn341 and Ser339 appears to stabilize the observed TP conformation (Supplementary Fig. 12b). In sum, TP binding in this DyP encapsulin system appears to be complex, relying on multiple types of interactions as well as general shape complementarity to facilitate cargo loading during shell self-assembly.

## The targeting peptide binding site is well-conserved in DyP encapsulin systems

Previous sequence analyses of TPs found in various classes of Family 1 encapsulin cargo proteins have highlighted substantial TP sequence conservation[15]. However, a similar analysis of the TP-binding site, located on the interior surface of the encapsulin shell protein P-domain, has not been carried out. To set our structural results regarding the KpDyP TP-shell interaction into context, we conducted a computational analysis across all DyP-containing encapsulins to assess the conservation of TP-binding site residues using the ConSurf server[57–59]. 1271 unique DyP encapsulin shell protein sequences were used for this analysis. The conservation of shell residues within 5 Å of the KpDyP TP was calculated via ConSurf analysis with standard parameters (Supplementary Data 4). We found that 17 shell residues (81% of interacting residues) are "conserved" with a ConSurf score of 6 or higher, while three residues had an "average" conservation score of 5, with only one residue yielding a "variable" score of 4. In particular, residues Arg34 and Asp38, involved in multiple H-bonding and ionic interactions discussed above, seem to be highly conserved. This analysis indicates that the TP binding site is substantially conserved across DyP encapsulin systems (Fig. 4d, e and Supplementary Data 5). Our results imply that the TP binding mode elucidated in this study is likely conserved for the majority of DyP encapsulins, the most numerous class of encapsulin systems.

## Mutational analysis of the KpDyP TP reveals a multifaceted binding mode as the basis for cargo encapsulation

To investigate the importance of individual TP residues and certain specific residue combinations, as well as TP-binding site residues of the KpEnc protomer, for mediating KpDyP encapsulation, a systematic mutational analysis via alanine scan was carried out. The conserved TP residues identified in our structure – GSLNIGSLK – were individually mutated to alanine to analyze their contribution to the overall cargo-loading process. In addition, Lys337 was also mutated individually. Further, 10 combinations of residues were mutated as well for a total of 20 TP mutants (Fig. 5a, b). To test the importance and cooperativity of hydrophobic interactions for TP binding, all combinations of the highly conserved residues Leu340, Ile342, and Leu345 were mutated. The well-conserved basic residue at the end of the TP, Lys336, and the less conserved Lys337, were also mutated in tandem to investigate the importance of ionic interactions for TP binding. Two well-conserved glycine residues, Gly338 and Gly343, hypothesized to be important for TP flexibility and shape complementarity, were also mutated simultaneously. Finally, all combinations of three mostly conserved residues potentially involved in hydrogen bonding interactions – Ser339, Ser344, and Asn341 – were mutated as well. These 20 different TP mutants were designed to specifically probe the hydrophobic, ionic, shape complementarity, and H-bonding contributions to TP binding. In addition to TP residues, four KpEnc protomer residues – Arg34, Asp38, Asp229, and Ile230 – were mutated as well (Fig. 5c). These four residues were identified in our structural analysis as likely important for mediating TP binding. To carry out these alanine scans, our established SUMO-TP_Enc setup was used. This system – relying on the small and monomeric SUMO protein as a cargo – was chosen to

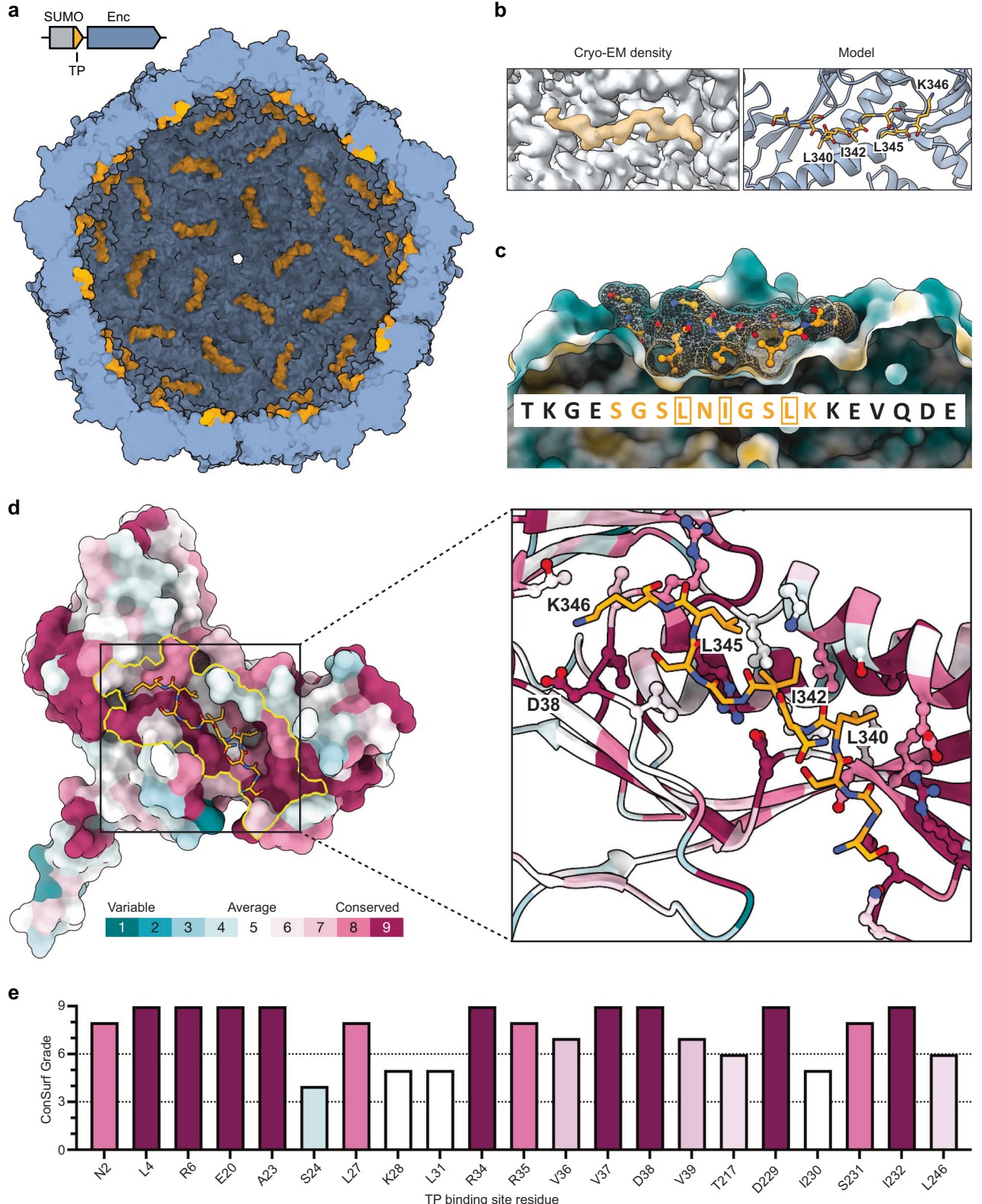

minimize the influence of steric hindrance – caused by the size or oligomerization state of the cargo – on cargo loading[56].

Each of the 24 mutant SUMO-TP_Enc systems, as well as a SUMO-TP control construct, were individually expressed and purified. SDS-PAGE analysis in combination with gel densitometry was then used to assess the amount of SUMO cargo loading (Fig. 5d, e and Supplementary Fig. 13). All samples were normalized based on the amount of

encapsulin shell protein. We found that substitution of any of the conserved hydrophobic residues (Leu340, Ile342, and Leu345), either individually or in combination (mutants 1–7), completely eliminated cargo encapsulation. Likewise, mutation of the C-terminal basic residue Lys346 alone or in combination (mutants 8 and 19) also abolished encapsulation while Lys347 had little impact on cargo loading (mutant 20). The G338A mutation (mutant 9) showed substantially decreased

**Fig. 4 | Structural and computational characterization of the TP-shell interaction. a** Cutaway view of a KpEnc encapsulin model – obtained from a SUMO-TP_Enc sample – in surface representation (blue) highlighting bound SGSLNIGSLK TPs (orange). **b** Cryo-EM density highlighting the TP binding site (left). The KpEnc encapsulin interior surface (gray) along with density observed for the TP (orange) are shown. Corresponding structural model of the TP binding site (right). The KpEnc protomer (blue, ribbon) and TP (orange, stick) are shown. **c** Side view of the TP binding site with the interacting TP residues – SGSLNIGSLK – shown in mesh and ball and stick model representation. The shape complementarity and hydrophobic pocket of the encapsulin shell protein (hydrophobic surface representation) and

the three hydrophobic TP residues L340, I342, and L345 (yellow boxes) are highlighted. **d** KpEnc protomer in surface representation colored by residue conservation as calculated via ConSurf analysis (left). The yellow outline indicates all residues within 5 Å of the bound TP (orange, stick representation). Magnified view of the TP binding site (right). Encapsulin shell protein residues within 5 Å of the TP are shown in ball and stick representation. Key hydrophobic and ionic interactions are highlighted. **e** ConSurf scores for the TP binding site residues of the encapsulin shell protein within 5 Å of the bound TP. Source data are provided as a Source Data file.

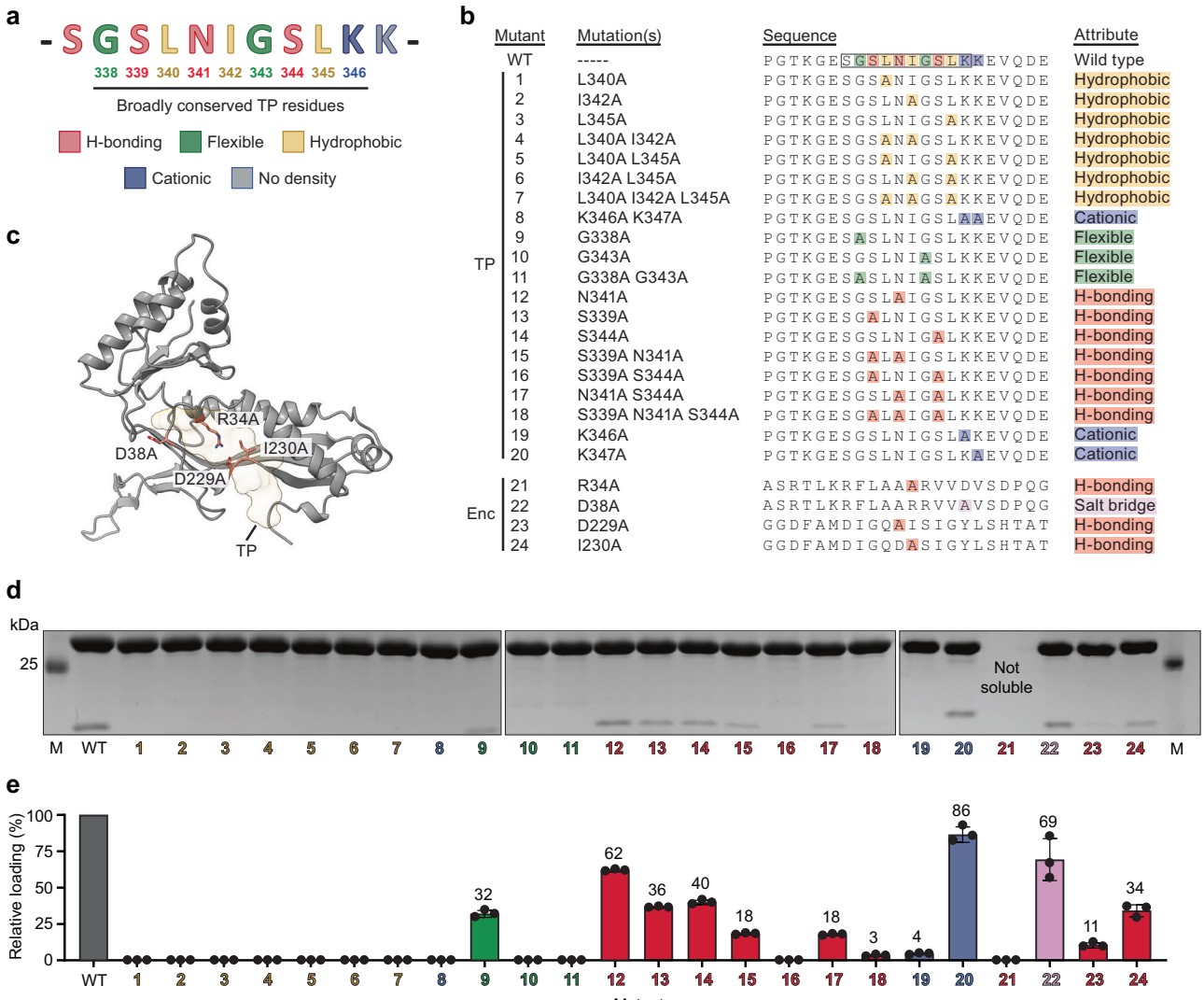

**Fig. 5 | Targeting peptide mutational analysis. a** TP sequence highlighting all residues subjected to mutational analysis (underlined) colored by the specific characteristics defined below. **b** Overview of all TP and KpEnc protomer mutants created. Residues are colored as in (**a**) (yellow, hydrophobic; blue, cationic; green, flexible; red, hydrogen bonding). TP residues with resolved cryo-EM density are highlighted in the wild-type sequence (WT; black outline). KpEnc protomer mutants (Enc) shown at the bottom. **c** KpEnc protomer viewed from the shell interior highlighting TP-binding site mutations. The TP is shown as semi-

transparent yellow surface. **d** Representative SDS-PAGE gels of purified cargo-loaded encapsulins for the WT and respective mutants showing KpEnc (upper band) and the co-purified encapsulated SUMO-TP mutant cargo (lower band). Source data are provided as a Source Data file. **e** Loading percentages of all SUMO-TP mutants as determined by gel densitometry. Loading percentage was normalized to WT loading. Data are shown as mean values. Error bars represent standard deviation of three independent experiments. Numbering and coloring are the same as in the preceding panels. Source data are provided as a Source Data file.

cargo loading, while any mutation including G343A (mutants 10 and 11) eliminated encapsulation altogether. One double Ser mutant (S339A with S344A; mutant 16) also resulted in a complete loss of cargo encapsulation. The other hydrogen bonding mutants (mutants 12–15, 17, and 18) all showed varying degrees of decreased encapsulation, with the triple mutant (S339A, N341A, S344A; mutant 18) resulting in

almost complete loss of cargo loading. The KpEnc protomer mutants showed substantial differences in their influence on cargo loading with mutants 21 and 23 almost completely abolishing loading while mutants 22 and 24 had less pronounced but still negative effects on cargo-loading efficiency. The residues altered in mutants 21 and 23 (Arg34 and Asp229) are involved in multiple H-bonding interaction with the

TP backbone and are clearly crucial for mediating TP binding. Asp38 (mutant 22), involved in a salt bridge with the TP residue Lys346, appears to be mostly dispensable with KpEnc residues Thr217 potentially able to form an alternative interaction with Lys346. Finally, mutant 24 confirms that the KpEnc residue Ile230 is important for mediating TP binding, likely via multiple H-bonding interaction with the TP backbone.

## Discussion

In this study, we elucidate the structural basis for DyP cargo encapsulation inside a Family 1 encapsulin nanocompartment found in the enterobacterial pathogen *K. pneumoniae*. DyP encapsulin systems represent the most prevalent class of encapsulin nanocompartments, with thousands of putative operons having been computationally identified, predominantly in the phyla Actinobacteria and Proteobacteria[26]. DyP encapsulins are found in many model organisms and important Gram-positive and Gram-negative human pathogens, including *Mycobacterium tuberculosis*[39] and the enterobacterial genera *Escherichia*, *Shigella*, and *Salmonella*. As both the respective shell protein and DyP cargo protein – including the TP – show high sequence homology across all computationally identified systems, our analysis is likely relevant for the majority of DyP encapsulins. Due to the large size of hexameric DyP cargo protein complexes, cargo loading in DyP encapsulins is generally fairly low, with only one to maximally three hexamers found per shell[21]. This leads to low TP occupancy making it difficult to visualize TP binding via cryo-EM. To overcome this challenge, we utilize a heterologous cargo-loading strategy based on a small and monomeric cargo protein (SUMO)[56], fused to the native KpDyP TP, which allowed us to achieve sufficient TP occupancy for cryo-EM analysis of the TP binding site. We find that DyP encapsulation is driven by specific interactions of the TP with a binding site located on the interior surface of the encapsulin shell protein. TP binding is facilitated by cooperative hydrophobic and ionic interactions, shape complementarity, and hydrogen bonding. Compared with previously structurally characterized TP-shell interactions (*Thermotoga maritima*, Flp-TP: GGDLGIRK; *Haliangium ochraceum*, Flp-TP: GSLGIGSLR; *Myxococcus xanthus*, Flp-TPs: SHPLTVGSLRR, PEKRLTVGSLRR; *Quasibacillus thermotolerans*, IMEF-TP: TVGSLIQ)[17,18,24,30] – all from ferroxidase cargos – the TP-shell interaction in our DyP system is mediated by three (Leu340, Ile342, and Leu345) instead of two hydrophobic residues. These three residues are accommodated by three distinct hydrophobic pockets located in the TP binding site with a defined register as highlighted by our mutational studies. Our high-resolution structure further allowed us to define H-bonding and ionic interactions, not previously described, also contributing to TP binding. Specifically, the C-terminal lysine (Lys346) forms a salt bridge with the conserved TP-binding site residue Asp38 while the binding site residues Arg34 and Asp229 form hydrogen bonds with the TP backbone. A unique feature of the *T. maritima* TP – an intra-TP salt bridge between its aspartate and arginine residues, is not observed in our or any other TP structure. However, our DyP TP exhibits an intramolecular hydrogen bond between residues Asn341 and Ser339 which, in analogy to the salt bridge observed in the *T. maritima* system, appears to stabilize the observed TP conformation. Our extensive mutational TP analysis illustrates that cargo loading is largely intolerant to TP mutation, with minor changes often leading to a complete loss of cargo encapsulation. These experimental results provide an explanation for the observed high degree of TP sequence conservation in DyP cargos specifically, and Family 1 encapsulin cargos more broadly. Moreover, our analysis of the TP binding site highlights a high degree of conservation for key shell protein residues involved in H-bonding, ionic, and hydrophobic TP interactions. To further investigate the contribution of the observed KpEnc shell protein residues toward TP binding, we created four KpEnc protomer mutants. Three of them, Arg34, Asp229, and Ile230,

were found to be highly important for mediating TP binding through primarily H-bonding interaction with the TP backbone, thus properly positioning the TP in the binding site. It is likely that the TP binding mode elucidated in this study represents a general and conserved feature of most DyP encapsulin systems.

It has been well established that DyPs are able to reduce hydrogen peroxide to water while concomitantly oxidizing a broad variety of co-substrates[40]. DyP substrates successfully tested to date include a variety of aromatic azo and anthraquinone dyes[60–62], lignin[40], guaiacol[60], β-carotene[63], and aromatic sulfides[64]. However, no native DyP co-substrates have been conclusively identified. This fact makes predicting the physiological function of DyPs in general, and DyP encapsulin systems in particular, difficult. It was recently reported that a DyP encapsulin in *M. tuberculosis* seems to increase bacterial resistance against oxidative stress in an infection model, specifically at low pH[39]. It was further hypothesized that DyP encapsulins may be required to remove toxic lipid hydroperoxides – in lieu of or in addition to hydrogen peroxide – created in the oxidative phagosomal environment encountered by *M. tuberculosis* during infection. It seems reasonable to hypothesize a similar role in stress response for the KpDyP_Enc system studied here. This hypothesis is supported by the fact that homologs of the two additional highly conserved genes found up- and downstream of the two-gene DyP encapsulin operon – encoding a formate dehydrogenase (FDH) and an RpoH-like sigma factor – have both been implicated in stress response functions. While RpoH-like sigma factors have been studied in multiple bacterial species as part of the heat-shock stress response[46], enterobacterial FDH has been shown to provide a fitness advantage in murine models of colitis, by providing additional reducing equivalents for respiration via formate oxidation[45]. Further, the fact that a conserved set of four genes – encoding FDH, KpDyP, KpEnc, and an RpoH-like sigma factor – are almost always found within mobile genetic elements suggests frequent horizontal transfer and a potential beneficial effect of acquiring this set of genes. It is possible that the two enzymatic activities found on these mobile genetic elements – formate oxidation and peroxide reduction – may generate a coordinated beneficial phenotype, resulting in an overall fitness advantage under formate- and peroxide-rich environments, for example during intestinal inflammation[65,66].

Here, we have elucidated the molecular and structural basis for DyP encapsulation. However, the functional roles such encapsulation may serve remain unclear. Encapsulin systems involved in iron storage and sulfur metabolism have been shown to directly depend on cargo enzyme encapsulation for optimal function[17,19,20]. Both systems store important nutrients – iron and sulfur, respectively – and stable sequestration inside a protein shell prevents toxicity and loss of stored nutrients. For DyP systems, a number of possible benefits of encapsulating a peroxidase activity have been proposed. For example, the shell could act as a molecular sieve to select for a certain type of substrate, thus counteracting the intrinsic promiscuity of DyP enzymes[7]. The protein shell could also provide a nanosized reaction chamber for optimizing DyP activity or increasing DyP stability or protease resistance[21,22,39,51]. As the native substrates of DyPs are currently unknown, it is difficult to test any of these hypotheses. Our biophysical analysis of free KpDyP showed that in addition to the active hexamer, other multimeric DyP states could be detected. Based on this observation, one additional function of encapsulation may be the stabilization of the most catalytically active oligomerization state of DyP.

Finally, by elucidating the structural basis of DyP cargo encapsulation and determining the contribution of individual TP residues toward cargo loading, we provide useful molecular-level detail for rationally modulating TP occupancy in future encapsulin engineering projects. This ability may prove crucial for fine-tuning cargo loading and optimizing catalytic function for a broad range of biomedical and biotechnological applications of encapsulin systems.

## Methods

### Chemicals and biological materials

All chemicals were used as supplied by vendors without further purification. Imidazole, Invitrogen Novex WedgeWell 14% tris-glycine Mini Protein Gels, NativePAGE 4 to 16% bis-tris Mini Protein Gels, Native-Mark Unstained Protein Standard, Isopropy-b-D-thiogalactopyranoside (IPTG), lysozyme, Spectra Multicolor Broad Range Protein Ladder, Pierce BCA protein assay kit, Tris base, Tris HCl, all restriction enzymes, and all cell culture media and reagents were purchased from Fisher Scientific, Inc. (USA). Gibson Assembly Master Mix and Q5 High-Fidelity DNA Polymerase were purchased from NEB (USA). Amicon Ultra-0.5 mL centrifugal units, Benzonase nuclease, and BL21 (DE3) Electrocompetent Cells used for *E. coli* expression were purchased from MilliporeSigma (USA). Ni-NTA agarose from Gold Biotechnology, Inc. (USA) was used for His-tagged protein purification.

### Instrumentation

Cell lysis was conducted via sonication with a Model 120 Sonic Dismembrator from Fisher Scientific, Inc. (USA). Proteins were quantified on a Nanodrop Spectrophotometer from ThermoFisher Scientific, Inc. (USA). Protein purification was carried out on an AKTA Pure fast liquid protein chromatography (FPLC) system; size exclusion chromatography (SEC) was carried out with a HiPrep 16/60 Sephacryl S-500 HR, Superose 6 10/300 GL, and Superdex 200 Increase 10/300 GL columns (Cytiva, USA); anion exchange was carried out with a HiTrap Q FF column (Cytiva, USA). Polyacrylamide gel electrophoresis (PAGE) and NativePAGE were performed in an XCell SureLock from Invitrogen/ThermoFisher Scientific (USA). Gel images were captured using a ChemiDoc Imaging System from Bio-Rad Laboratories, Inc. (USA). DLS was carried out on an Uncle from Unchained Labs (USA). TEM was carried out on a Morgagni 100 keV Transmission Electron Microscope (FEI, USA). Plate-based assays were conducted on the Synergy H1 Microplate Reader from BioTek Instruments (USA). EM grid glow discharging was conducted with a PELCO easiGlow system by Ted Pella, Inc. (USA). A Talos Arctica Cryo Transmission Electron Microscope by ThermoScientific, Inc. (USA) equipped with a K2 Summit direct electron detector by Gatan, Inc. (USA) located at the University of Michigan Life Sciences Institute was used for cryo-EM. Other materials are listed along with corresponding methods below.

### Software

The following software was used throughout this work: cryoSPARC v4.1.2[67] (cryo-electron microscopy), Fiji/ImageJ v2.1.0/1.53c[68] (densitometric data analysis and TEM images), GraphPad Prism for Mac OS v10.0.0 (chromatography, kinetic, conservation score, and kinetic graphs), Bio-Rad Image Lab Touch Software (gel imaging), Coot v9.8.1[69] and Phenix v1.20.1-4487-000[70] (model building), UCSF Chimera v1.16[71] and ChimeraX v1.16.1[72] (cryo-EM density and model visualization), and UNICORN 7 (FPLC system control and chromatography). Online software suites or tools are listed along with corresponding methods below.

### Bioinformatic analyses

Phylogenetic analyses were initiated using a representative encapsulin from *K. pneumoniae* (UniParc ID: UPI001261BC85) to generate an NCBI identical proteins group (Supplementary Data 1). Representative sequences from each resulting genus (Supplementary Data 2) were then used to create a phylogenetic analysis via standard programs and workflow from NGphylogeny.fr[73]. For DyP analyses, a curated list of Family 1 encapsulins[74] was sorted according to cargo type, resulting in six major classes. Only the resulting DyP systems were used and their C-termini were aligned via Clustal Omega 1.2.3[75] with 20 residues centered on the consensus peak or, when limited by sequence length, using the last 20 C-terminal residues. Unique DyP encapsulin shell proteins from the curated Family 1 encapsulins list were then analyzed

via ConSurf using default settings to determine the evolutionary conservation of KpEnc (Supplementary Data 3)[57–59].

### Protein production

For all target proteins other than the SUMO-TP_KpEnc mutants, plasmids were constructed using *E. coli* codon-optimized gBlock genes, synthesized by IDT (USA), inserted into the pETDuet-1 vector via Gibson assembly using the NdeI and PacI restriction sites (Supplementary Table 2). *E. coli* BL21 (DE3) was transformed with the respective plasmids via electroporation and 25% glycerol bacterial stocks were prepared and stored at −80 °C until further use. Starter cultures were grown in 5 mL LB with 100 mg/mL ampicillin at 37 °C overnight. For all constructs, 500 mL of LB with ampicillin was inoculated with overnight starter cultures and grown at 37 °C to an OD600 of 0.4–0.5, then induced with 0.1 mM IPTG and grown further at 30 °C overnight for ca. 18 h. Constructs containing DyP were also supplemented with 0.3 mM 5-aminolevulinic acid hydrochloride and 100 µM FeSO4 at time of induction. Cells were harvested via centrifugation at 10,000 rcf for 15 min at 4 °C and pellets were frozen and stored at −80 °C until further use. SUMO-TP_KpEnc mutants were constructed via standard Quik-Change Site-Directed mutagenesis as developed by Stratagene (La Jolla, CA) per standard protocol. Mutant constructs were expressed as above.

### Protein purification

For all proteins other than the His-tagged KpDyP and SUMO-TP_Enc samples, frozen bacterial pellets were thawed on ice and resuspended in 5 mL/g (wet cell mass) of cold Tris Buffered Saline (20 mM Tris pH 7.5, 150 mM NaCl). Lysis components were added (0.5 mg/mL lysozyme, 1 mM tris(2-carboxyethyl)phosphine [TCEP], one SIGMA*FAST* EDTA-free protease inhibitor cocktail tablet per 100 mL, 0.5 mM MgCl2, and 25 units/mL Benzonase nuclease) and samples were lysed on ice for 10 min. Samples were then sonicated at 60% amplitude for 5 min total (eight seconds on, 16 s off) until no longer viscous. After sonication, samples were centrifuged at 8000 rcf for 15 min at 4 °C. Samples were then subjected to 10% polyethylene glycol (PEG) 8000 precipitation (lysate brought to 10% PEG 8 K and 500 mM NaCl and incubated for 30 min on ice, then centrifuged 8000 rcf for 15 min). Supernatant was discarded and the pellet was resuspended in 5 mL TBS pH 7.5 and filtered using a 0.22 µm syringe filter (Corning, USA). The protein sample was then loaded on an AKTA Pure and purified via a Sephacryl S-500 column. Sample fractions were pooled, and buffer exchanged using Amicon Ultra Centrifugal Filter units (MW cutoff 30 kDa, Millipore) into AIEX Buffer (20 mM Tris pH 7.5) and loaded onto an AKTA Pure, then purified via HiTrap Q-FF by linear gradient into AIEX Buffer with 1 M NaCl. Sample flow-through was collected and centrifuged at 10,000 rcf for 10 min, then loaded on an AKTA Pure for final purification via a Superose 6 10/300 GL column pre-equilibrated with TBS pH 7.5.

For free His-tagged KpDyP purification, the sample was lysed as above in NTA Resuspension Buffer (50 mM Tris pH 7.5, 150 mM NaCl, 10 mM imidazole, and 5% glycerol). Lysate was bound to Ni-NTA resin pre-equilibrated with NTA Resuspension Buffer via rocking at 4 °C for 45 min. Supernatant was discarded and the bound sample was washed once with NTA Resuspension Buffer and a second time with NTA Resuspension Buffer with 20 mM imidazole. Free His-tagged KpDyP was then eluted three times with Elution Buffer (50 mM Tris pH 7.5, 150 mM NaCl, 350 mM imidazole, 5% glycerol) and frozen at −80 °C for future use.

For SUMO-TP_KpEnc, frozen bacterial pellets were thawed on ice and resuspended in 5 mL/g (wet cell mass) of cold Tris Buffered Saline (20 mM Tris pH 7.5, 150 mM NaCl). Lysis components were added (0.5 mg/mL lysozyme, one SIGMA*FAST* EDTA-free protease inhibitor cocktail tablet per 100 mL, 2 mM MgCl2, and 50 units/mL Benzonase nuclease) and samples were lysed on ice for 10 min. Samples were then

sonicated at 70% amplitude for 3 min total (10 s on, 20 s off) until no longer viscous. After sonication, samples were centrifuged at 21,000 rcf for 10 min at 4 °C. Supernatants were then heated to 60 °C for 20 min, then centrifuged again as before. Clarified supernatant was collected and saturated with 50% $(NH_4)_2SO_4$ (0.314 g/mL) and incubated overnight while rocking at 4 °C. After 18 h, samples were centrifuged at 8000 rcf, then resuspended in 10 mL buffer containing 20 mM Tris pH 7.5 and 150 mM NaCl. Samples were then concentrated to 500 µL using a 15 mL 100 kDa MWCO Amicon centrifugal unit, then centrifuged at 10,000 rcf for 10 min, and loaded on an AKTA Pure for final purification via a Superose 6 10/300 GL column pre-equilibrated with TBS pH 7.5. All proteins were stored at 4 °C until use. Proteins were quantified via Pierce BCA protein assay kit (Fisher Scientific, Inc.; USA).

## Negative stain transmission electron microscopy (TEM)
All samples were diluted to 0.1–0.2 mg/mL in TBS pH 7.5 and immediately stained and imaged. Negative stain TEM was carried out on the various samples with 200-mesh gold grids coated with extra thick (25–50 nm) formvar-carbon film (EMS, USA) made hydrophilic by glow discharging at 5 mA for 60 s. Briefly, 3.5 µL of sample was added to the grid and incubated for 30 s, wicked with filter paper, and washed once with distilled water and once with 0.75% (w/v) uranyl formate before staining with 8.5 µL of uranyl formate for 30 s. TEM images were captured using a Morgagni transmission electron microscope at 100 keV at the University of Michigan Life Sciences Institute.

## Peroxidase activity assays
Free and encapsulated DyP peroxidase (KpDyP and KpDyP_Enc) were purified and diluted to equimolar heme concentrations as determined via heme assay kit (Sigma-Aldrich, USA) per manufacturer protocol to analyze enzyme activity via ABTS assays. Reactions contained 50 mM 2-(N-morpholino)ethanesulfonic acid (MES) buffer and 150 mM NaCl at pH 5.5, with 5 mM ABTS and each respective enzyme (free or encapsulated) standardized to 600 nM heme concentration. Reactions were initiated by the addition of varying concentrations of hydrogen peroxide ($H_2O_2$) (two-fold dilutions for final reaction concentrations ranging from 10 mM to 1.2 µM). The oxidation of ABTS was measured at 420 nm in a BioTek Synergy H1 microplate reader at a final volume of 100 µL in Corning 96-well flat clear bottom black polystyrene microplates in 10 s intervals for a total of 15 min to evaluate enzyme activity and the rate of reaction. All assays were conducted in triplicate. Non-linear regression curve analysis with Michaelis-Menten fit of initial velocities was conducted with Graph-Pad Prism 10.

## Dynamic and static light scattering analyses (DLS and SLS)
All sizing and polydispersity measurements were carried out on an Uncle by Unchained Labs (USA) at 30 °C in triplicate. Purified samples were adjusted to 0.2–0.4 mg/mL of monomer in the standard TBS pH 7.5 buffer and centrifuged at 10,000 rcf for 10 min, then immediately analyzed at 25 °C via DLS (Supplementary Figs. 2 and 3).

## Analytical size exclusion and native PAGE analysis of KpDyP
Purified free KpDyP was subjected to size exclusion analysis on a Superdex 200 Increase 10/300 GL column. Fractions were then analyzed via native PAGE in an Invitrogen XCell SureLock using NativePAGE 3–12% bis-tris mini protein gels, 1X NativePAGE Running Buffer, and NativeMark Unstained Protein Standard from Fisher Scientific (USA) with 1× running buffer made from 10× Tris/Glycine Buffer from Bio-Rad Laboratories, Inc. (USA). Approximately 10 µg of protein was loaded per well. NativePAGE gels were run at 150 V for 1 h, then 250 V for 2.5 h at 4 °C. Gels were stained with ReadyBlue Protein Gel Stain from Sigma-Aldrich (USA) and imaged and analyzed on a ChemiDoc Imaging System by Bio-Rad Laboratories, Inc. (USA).

## Cryo-electron microscopy (cryo-EM)
**Sample preparation.** The purified KpDyP_Enc and SUMO-TP_Enc encapsulin samples were concentrated to 3 mg/mL and the free KpDyP sample was diluted to 0.45 mg/mL in 150 mM NaCl, 20 mM Tris pH 7.5. 3.5 µL of protein samples were applied to freshly glow discharged Quantifoil R1.2/1.3 Cu 200-mesh grids for the KpDyP_Enc and SUMO-TP_Enc samples or Quantifoil R2/1 Cu 200-mesh grids for the KpDyP sample and plunged into liquid ethane using an FEI Vitrobot Mark IV (KpDyP_Enc and SUMO-TP_Enc: 100% humidity, 22 °C, blot force 20, blot time 4 s, drain time 0 s, wait time 0 s; KpDyP: 100% humidity, 22 °C, blot force 5, blot time 2 s, drain time 0 s). The frozen grids were clipped and stored in liquid nitrogen until data collection.

**Data collection.** Cryo-electron microscopy movies for the KpDyP_Enc and SUMO-TP_Enc samples were collected using a ThermoFisher Scientific Talos Arctica operating at 200 kV with a Gatan K2 Summit direct electron detector. Movies were collected at 45,000× magnification using the Leginon[76] software package with a pixel size of 0.91 Å/pixel (Supplementary Table 1).

KpDyP movies were collected using a ThermoFisher Scientific Titan Krios G4i operating at 300 kV equipped with a Gatan K3 direct electron detector with a Bioquantum imaging filter. KpDyP movies were collected at 105,000× magnification using the SerialEM with a pixel size of 0.832 Å/pixel (Supplementary Table 1).

**Data processing.** CryoSPARC v4.1.2 was used to process the KpDyP_Enc dataset[67]. For the KpDyP_Enc dataset, 1183 movies were imported, motion corrected using patch motion correction, and the CTF-fit was estimated using patch CTF estimation. Exposures with CTF fits worse than 5 Å were discarded from the dataset, resulting in 1011 remaining movies. 232 particles were manually picked to create a template for particle picking. Template picker was used to identify particles and 75,967 particles were extracted with a box size of 384 pixels. Two rounds of 2D classification yielded 67,397 particles. The particles were then downsampled to a box size of 128 pixels and classified using ab-initio reconstruction with 6 classes and I symmetry. 64,969 particles contained in the two nearly identical classes were then used for homogeneous refinement against the ab-initio map with I symmetry imposed, per-particle defocus optimization, per-group CTF parameterization, and Ewald sphere correction enabled, resulting in a 2.52 Å resolution map. To visualize the encapsulated DyP, the shell was removed via particle subtraction using a static mask of the shell. The shell-subtracted particles were then sorted by 2D classification to identify particles that contained defined internal densities corresponding to DyP complexes.

CryoSPARC v4.4.1+231114 was used to process the KpDyP dataset. For KpDyP, 2468 movies were imported into CryoSPARC Live which was used to perform patch motion correction and patch CTF fit estimation. Movies with CTF fits worse than 5 Å were discarded, resulting in 2453 remaining movies. Blob picker was used in CryoSPARC Live to pick initial particles with a minimum particle diameter of 100 Å and a maximum diameter of 180 Å. Picked particles were then sorted in CryoSPARC Live using 2D classification resulting in 681,023 selected particles that were then used to generate templates for template-based particle picking. CryoSPARC was used for all further processing steps. 2,603,093 particles were picked using template picker and were extracted with a box size of 256 pixels. Two rounds of 2D classification resulted in 950,679 selected particles. The particles were further classified using ab-initio reconstruction with 3 classes and D3 symmetry imposed. The major ab-initio class contained 601,127 particles, which resulted in a 2.57 Å resolution map when reconstructed using homogeneous refinement with D3 symmetry imposed, per-particle scale minimization, per-particle defocus optimization, per-group CTF parameterization, and Ewald sphere correction enabled. Heterogeneous refinement with three classes and D3 symmetry imposed against the 2.57 Å map yielded a majority class containing 469,352 particles. These

particles were used in a subsequent homogeneous refinement job using the same parameters as above, yielding a 2.61 Å resolution map. Heterogeneous refinement was repeated against the 2.61 Å resolution map, resulting in a majority class containing 431,317 particles. Homogeneous refinement of these particles using the same parameters as previously resulted in an improved 2.52 Å resolution map. The particles were then polished using reference-free motion correction and were used as an input for another homogeneous reconstruction job using the same parameters as earlier, resulting in a 2.48 Å resolution map. The map quality was further improved by performing a local refinement with D3 symmetry and force re-do GS split applied, resulting in a final 2.39 Å resolution map.

CryoSPARC v4.1.2 was used to process the SUMO-Tp_Enc dataset. or SUMO-TP_Enc, 1654 movies were imported into CryoSPARC, motion corrected using patch motion correction, and CTF fit was estimated using patch CTF estimation. Movies with CTF fits worse than 5 Å were discarded, leaving 1531 movies for particle picking. 200 particles were picked manually and used to generate templates for particle picking. Template picker was used to pick particles. 115,682 particles with a box size of 384 pixels were then extracted. Good particles were further selected by three iterations of 2D classification, resulting in 102,513 particles, which were used in a six class ab-initio run. The major ab-initio class contained 101,111 particles, which were used as an input for homogeneous refinement against the ab-initio map with I symmetry imposed, per-particle defocus optimization, per-group CTF parameter optimization, and Ewald sphere correction, which resulted in a final 2.41 Å map.

**Model building.** For building the KpDyP_Enc shell model, a protomer of the homologous encapsulin from *Mycobacterium smegmatis* (PDB: 7BOJ) was manually placed into the cryo-EM map using ChimeraX v.1.2.5, then fit using the fit in map command[21,77]. The model was then manually mutated and refined using Coot v9.8.1 until it was deemed a satisfactory fit[69]. Real-space refinement against the map was performed using Phenix v1.20.1-4487-000 with three macrocycles, minimization_global enabled, local_grid_search enabled, and adp refinement enabled[78]. The symmetry operators were identified from the map using the map_symmetry command and applied using apply_ncs to generate an icosahedral shell with 60 copies of the asymmetric unit. The NCS-expanded shell was then refined again using real-space refinement with three macrocycles, minimization_global enabled, local_grid_search enabled, adp refinement enabled, and NCS constraints enabled. BIOMT operators were found using the find_ncs command, and manually placed in the headers of the .pdb file. Model building for SUMO-TP_Enc followed a nearly identical strategy as for the KpDyP_Enc model, except that the KpDyP_Enc model was used as a model for initial placement.

For building the KpDyP model, an AlphaFill model (AF-A0A3Z8UGY6-F1) containing heme in the active cite was docked using ChimeraX[79]. NCS operators were found from the map using map_symmetry in Phenix and applied using apply_ncs to generate the D3 complex containing six monomers of KpDyP. The model was refined against the map using real-space refine with NCS constraints, morphing, and all other settings set to default. BIOMT operators were found using the find_ncs command and manually placed in the headers of the .pdb file. The models and cryo-EM densities for KpDyP_Enc, KpDyP, and SUMO-TP_Enc were deposited in the Protein Data Bank (PDB) and the Electron Microscopy Data Bank (EMDB) under the PDB IDs 8U50, 8U4Z, and 8U51, and EMDB IDs EMD-41905, EMD-41904, and EMD-41906, respectively.

**Analyses of relative TP mutant cargo loading**
The purified SUMO-TP_Enc TP mutants were concentrated to equimolar concentrations based on shell protein amount via gel densitometry and analyzed by SDS-PAGE. Then, the relative amounts of

encapsulated SUMO-TP mutants were determined via gel densitometry using Fiji/ImageJ and compared to the amount encapsulated in the wild-type sample (normalized as 100% loading)[68].

**Reporting summary**
Further information on research design is available in the Nature Portfolio Reporting Summary linked to this article.

## Data availability
Cryo-EM maps and structural models have been deposited in the Electron Microscopy Data Bank (EMDB) and the Protein Data Bank (PDB) and are publicly available. PDB IDs: 8U50, 8U4Z, and 8U51. EMDB IDs: EMD-41905, EMD-41904, and EMD-41906. Sequence and alignment data has been supplied as Supplementary Data. Source data are provided with this paper.

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

## Acknowledgements

We acknowledge funding from the NIH (R35GM133325). Research reported in this publication was supported by the University of Michigan Cryo-EM Facility (U-M Cryo-EM). U-M Cryo-EM is grateful for support from the U-M Life Sciences Institute and the U-M Biosciences Initiative. Molecular graphics and analyses performed with UCSF ChimeraX, developed by the Resource for Biocomputing, Visualization, and Informatics at the University of California, San Francisco, with support from the National Institutes of Health R01GM129325 and the Office of Cyber Infrastructure and Computational Biology, National Institute of Allergy and Infectious Diseases.

## Author contributions

J.A.J., M.P.A., and T.W.G. designed the project. J.A.J. and T.W.G. conducted the bioinformatic analyses. J.A.J. and M.P.A. conducted the laboratory experiments, with J.A.J. conducting the enzyme kinetic assays and negative stain transmission electron microscopy, while M.P.A. collected and analyzed the cryo-EM data. M.P.A. and T.W.G. processed the cryo-EM data. M.P.A. built the structural models. J.A.J. and T.W.G. wrote the manuscript. T.W.G. oversaw the project in its entirety.

## Competing interests

The authors declare no competing interests.
