## [Peer Review File · Nature Communications]

Structural basis for peroxidase encapsulation inside the encapsulin from the Gram-negative pathogen *Klebsiella pneumoniae*REVIEWER COMMENTS

Reviewer #1 (Remarks to the Author):

In this study, the authors examined an encapsulin containing DyP1 from the pathogen *Klebsiella pneumoniae*. Encapsulins are protein compartments in prokaryotes, serving roles in iron storage, sulfur metabolism, and responding to oxidative stress. Each subunit of the encapsulin shell contains a binding site for targeting peptides (TP) that helps with encapsulation. In this study, the authors have combined cryo-electron microscopy with TP mutagenesis to advance the knowledge on the molecular basis for cargo encapsulation. While there were earlier studies on encapsulin structures from other bacteria, like *Myxococcus* and *Mycobacterium*, this research, although well-executed, doesn't introduce major new findings because the basic features are similar to what was described before. The mutagenesis studies revealed that changing individual TP amino acids didn't significantly affect cargo loading. This is likely because DyP cargo generally has highly conserved TP sequences. In summary, the study doesn't provide significant new insights.

Reviewer #2 (Remarks to the Author):

In this comprehensive manuscript, Giessen and coworkers present cryo-EM structure and mutagenesis data depicting the interaction of a targeting peptide and its associated encapsulin shell from *Klebsiella pneumoniae*, which will be of high interest to the field. In addition to the targeting peptide/shell interaction results, they provide phylogenetic and bioinformatics data and analysis of peroxidase (DyP) encapsulins, with speculation as to their native function, and substantial biophysical and biochemical analysis of the *Klebsiella pneumoniae* DyP encapsulin system following heterologous production in *Escherichia coli*. While related, the function and assembly of encapsulins are tough to fit into a single manuscript while doing each aspect justice, and here I felt that there was only a shallow exploration of the function/bioinformatics portion of the work, and a deeper exploration of the targeting peptide/shell structure, leading to a somewhat imbalanced feel. I could see splitting it into two manuscripts yet understand that sometimes other factors go into such decisions. Either way, I recommend adding some context plus one more experiment to strengthen the support for the section on targeting peptide/encapsulin shell interacting residues, as described below:

1. The Giessen lab has worked on the interaction of targeting peptide and shell for some time, with a series of publications on different model systems, and this is the first to definitively demonstrate the importance of various interactions as well as shape in this interaction. While some context is given, this work would benefit from additional discussion of the literature, to better set up and explain the results and their novelty.

2. The mutagenesis work explores changes to the amino acids in the targeting peptide, but not the encapsulin shell. To truly be comprehensive, I would like to see an alanine scan or similar to the shell residues of the binding site as well – at least to the several key residues that were hypothesized from the structural data.

3. The alanine scan and combinatorial selections were well reasoned, with one exception. Why was the final lysine of the consensus motif, K346, not mutated to alanine without altering K347 as well?

Minor:

4. The methods mention “buffer exchange” in the protein purification section – please describe the method used.

5. It is odd to me that for the H-bonding residues, the triple mutant (18) has some loading while the double (16) does not. Some discussion is warranted as to why this might be (or is it reflective of the error of the experiment?).

Reviewer #3 (Remarks to the Author):

The manuscript by Jones et al. used cryo-EM imaging/reconstruction and a surrogate system to determine the structure of *Klebsiella pneumoniae* encapsulin and the binding site of its cargo-loading peptide. Mutagenesis studies were performed to pinpoint the residues in the peptide critical for binding. This is a solid work that details the structural basis of this particular encapsulin/cargo-loading peptide pair, which is likely shared across a wide range of encapsulin/cargo systems in many bacteria. However, similar structure and binding site information have already been reported in the original X-ray structure of *Thermotoga maritima* encapsulin and its cargo-loading peptide binding site (Sutter ... Ban (2008). Structural basis of enzyme encapsulation into a bacterial nanocompartment. *Nature Structural & Molecular Biology*, 15(9), 939–947). While this manuscript has cited this paper, it does not include any comparison and it does not discuss the common findings and new knowledge generated by the new study.

A few minor points:

Fig. 1d: What is the unlabeled dashed curve?

Have the authors tried to estimate the number of DyP hexamers in each Enc?

Fig. 2d: Why has DyP in Enc lost ~25% of heme while free DyP is nearly 100% loaded with heme? This seems in conflict with the statement “one additional function of encapsulation may be the stabilization of the most catalytically active oligomerization state of DyP” (p. 21, lines 474-6).

The cryoEM reconstruction of free DyP is only solved at 3.9 Å, a low-resolution structure for a hexamer of ~244 kDa protein complex with D3 symmetry. Sub-3 Å structure is expected with current cryo-EM technology.

Fig. 3f: How reliable are the 2D class averages shown in Fig. 3f? Have the authors tried to solve the 3D structure of the loaded DyP using the shell-subtracted images? A low-resolution 3D reconstruction should be possible if the 2D class averages are meaningful.

Reviewer #4 (Remarks to the Author):

The manuscript "Structural basis for peroxidase encapsulation in a protein nanocompartment" by Jones et al. describes the structural and molecular basis for the encapsulation of the dye-decolorizing peroxidase (DyP) hexameric complex inside the encapsulin nanocompartment from the Gram-negative pathogen *Klebsiella pneumoniae* (named KpEnc). Using cryo-EM, the authors determined the structure of the DyP-loaded KpEnc and the free DyP hexamer (expressed in *E. coli*) at 2.5 and 3.9 Å resolution, respectively. DyP encapsulation is mediated by a targeting motif (named TP), located at its C-terminus, which recognizes a specific Enc binding pocket. However, this interaction is unresolved in the native, icosahedral DyP-loaded KpEnc due to the low TP occupancy. This drawback is smartly solved using a chimeric, non-native cargo that includes the DyP TP. This small-size cargo is encapsulated with a high copy number, and the TP-pocket interaction is described in the KpEnc system at 2.4 Å resolution. An exhaustive TP mutagenesis analysis establishes with precision the role of each residue and/or group of similar residues toward cargo encapsulation.

As would be anticipated from this group, the work presented is of excellent quality and advances the structural description of encapsulins in an incremental manner. I recommend publication, but suggest that the following very minor issues be (optionally) addressed:

1. To highlight this work, I would change the title of the manuscript to "Structural basis for peroxidase encapsulation inside the encapsulin from the Gram-negative pathogen *Klebsiella pneumoniae*"
2. The descriptions of T1, T3 or T4 (l. 45) are not formal to refer to triangulation numbers; I would specify the first citation as a T=1 shell, indicating the adopted nomenclature of T1 and so on.
3. Following my general description above, and as described also by the authors in the abstract and at the end of the introduction, in my opinion, it would be more logic to show the DyP-loaded KpEnc and the free DyP hexamer structures in this order.

Point-by-point response to reviewer comments:

Author replies shown in blue

Yellow background indicates new text added to the manuscript

Reviewer #1 (Remarks to the Author):

In this study, the authors examined an encapsulin containing DyP1 from the pathogen *Klebsiella pneumoniae*. Encapsulins are protein compartments in prokaryotes, serving roles in iron storage, sulfur metabolism, and responding to oxidative stress. Each subunit of the encapsulin shell contains a binding site for targeting peptides (TP) that helps with encapsulation. In this study, the authors have combined cryo-electron microscopy with TP mutagenesis to advance the knowledge on the molecular basis for cargo encapsulation. While there were earlier studies on encapsulin structures from other bacteria, like *Myxococcus* and *Mycobacterium*, this research, although well-executed, doesn't introduce major new findings because the basic features are similar to what was described before.

The mutagenesis studies revealed that changing individual TP amino acids didn't significantly affect cargo loading. This is likely because DyP cargo generally has highly conserved TP sequences. In summary, the study doesn't provide significant new insights.

Author reply: We are afraid that Reviewer #1 is mistaken with this statement. In fact, exactly the opposite of what Reviewer #1 states is the case, namely, changing individual TP amino acids does severely influence cargo loading.

Reviewer #2 (Remarks to the Author):

In this comprehensive manuscript, Giessen and coworkers present cryo-EM structure and mutagenesis data depicting the interaction of a targeting peptide and its associated encapsulin shell from *Klebsiella pneumoniae*, which will be of high interest to the field. In addition to the targeting peptide/shell interaction results, they provide phylogenetic and bioinformatics data and analysis of peroxidase (DyP) encapsulins, with speculation as to their native function, and substantial biophysical and biochemical analysis of the *Klebsiella pneumoniae* DyP encapsulin system following heterologous production in *Escherichia coli*. While related, the function and assembly of encapsulins are tough to fit into a single manuscript while doing each aspect justice, and here I felt that there was only a shallow exploration of the function/bioinformatics portion of the work, and a deeper exploration of the targeting peptide/shell structure, leading to a somewhat imbalanced feel. I could see splitting it into two manuscripts yet understand that sometimes other factors go into such decisions. Either way, I recommend adding some context plus one more experiment to strengthen the support for the section on targeting peptide/encapsulin shell interacting residues, as described below:

1. The Giessen lab has worked on the interaction of targeting peptide and shell for some time, with a series of publications on different model systems, and this is the first to definitively demonstrate the importance of various interactions as well as shape in this interaction. While some context is given, this work would benefit from additional discussion of the literature, to better set up and explain the results and their novelty.

Author reply: We thank Reviewer #2 for this suggestion. We have expanded our literature discussion in the introduction section as requested.

Page 3, line 63 (introduction): "TPs are usually connected to the cargo protein by a flexible linker with high glycine and proline content which likely minimizes steric clashes between adjacent cargo proteins within the shell. So far, only TP-shell interactions of ferroxidase cargo proteins (ferritin-like proteins and iron-mineralizing encapsulin-associated firmicute cargos) have

been investigated structurally^{17,18,24,30}. It was found that TP binding sites always reside on the luminal surface of encapsulin protomers with between 7 and 12 TP residues tightly interacting with a narrow binding pocket. A binding mode primarily based on hydrophobic interactions has been proposed where two conserved hydrophobic side chains – generally Leu, Ile, or Val – interact with hydrophobic patches within the TP-binding site. However, no detailed experimental analysis of the contributions of individual TP or TP-binding site residues towards cargo encapsulation has been reported.”

2. The mutagenesis work explores changes to the amino acids in the targeting peptide, but not the encapsulin shell. To truly be comprehensive, I would like to see an alanine scan or similar to the shell residues of the binding site as well – at least to the several key residues that were hypothesized from the structural data.

3. The alanine scan and combinatorial selections were well reasoned, with one exception. Why was the final lysine of the consensus motif, K346, not mutated to alanine without altering K347 as well?

Author reply: We thank Reviewer #2 for these suggestions. We have created six new mutants (four shell mutants: R34A, D38A, D229A, I230A + two new TP mutants: K346A, K347A) to address the points raised above. The shell mutants are, as suggested by Reviewer #2, focused on the key shell residues involved in ionic and H-bonding interactions as identified through our structural analysis. We found that mutating, in particular, shell residues Arg34, Asp229, and Ile230 – suggested to mediate key H-bonding interactions with the TP backbone – had a profound effect on TP binding confirming our structural analysis. Further, the single Lys346 mutant was sufficient to abolish TP binding altogether with Lys347 having little effect, indicating that the salt bridge and hydrogen bond formed by Lys346 with the TP binding site is crucial for mediating cargo loading. We have updated Fig. 5 which now contains the results of all 24 mutants created and analyzed in this study, as well as Supplementary Fig. 13 with newly obtained mutant-related data. In addition, substantial changes were made to the relevant paragraphs in the results and discussion sections.

Page 17, line 376 (results): “To investigate the importance of individual TP residues and certain specific residue combinations, as well as TP-binding site residues of the KpEnc protomer, for mediating KpDyP encapsulation, a systematic mutational analysis via alanine scan was carried out.”

Page 17, line 379 (results): “In addition, Lys337 was also mutated individually. Further, 10 combinations of residues were mutated as well for a total of 20 TP mutants (Fig. 5a,b).”

Page 17, line 391 (results): “In addition to TP residues, four KpEnc protomer residues – Arg34, Asp38, Asp229, and Ile230 – were mutated as well (Fig. 5c). These four residues were identified in our structural analysis as likely important for mediating TP binding. To carry out these alanine scans, our established SUMO-TP_Enc setup was used.”

Page 19, line 414 (results): “Each of the 24 mutant SUMO-TP_Enc systems, as well as a SUMO-TP control construct, were individually expressed and purified.”

Page 19, line 420 (results): “Likewise, mutation of the C-terminal basic residue Lys346 alone or in combination (mutants 8 and 19) also abolished encapsulation while Lys347 had little impact on cargo loading (mutant 20).”

Page 19, line 428 (results): “The KpEnc protomer mutants showed substantial differences in their influence on cargo loading with mutants 21 and 23 almost completely abolishing loading while mutants 22 and 24 had less pronounced but still negative effects on cargo loading efficiency. The residues altered in mutants 21 and 23 (Arg34 and Asp229) are involved in multiple H-bonding interaction with the TP backbone and are clearly crucial for mediating TP binding. Asp38 (mutant 22), involved in a salt bridge with the TP residue Lys346, appears to be mostly dispensable with KpEnc residues Thr217 potentially able to form an alternative interaction with Lys346. Finally,

mutant 24 confirms that the KpEnc residue Ile230 is important for mediating TP binding, likely via multiple H-bonding interaction with the TP backbone.”

Page 21, line 481 (discussion): “To further investigate the contribution of the observed KpEnc shell protein residues toward TP binding, we created four KpEnc protomer mutants. Three of them, Arg34, Asp229, and Ile230, were found to be highly important for mediating TP binding through primarily H-bonding interaction with the TP backbone, thus properly positioning the TP in the binding site.”

Minor:

4. The methods mention “buffer exchange” in the protein purification section – please describe the method used.

Author reply: We have updated the methods section to now detail how the buffer exchanges were accomplished.

Page 25, line 626: “Sample fractions were pooled, and buffer exchanged using Amicon Ultra Centrifugal Filter units (MW cutoff 30 kDa, Millipore) into AIEX Buffer (20 mM Tris pH 7.5) and loaded onto an AKTA Pure, then purified via HiTrap Q-FF by linear gradient into AIEX Buffer with 1M NaCl.”

5. It is odd to me that for the H-bonding residues, the triple mutant (18) has some loading while the double (16) does not. Some discussion is warranted as to why this might be (or is it reflective of the error of the experiment?).

Author reply: Reviewer #2 is correct in that this is likely caused by experimental variability. The largest observed standard deviation based on three independent purification experiments is larger than the average value measured for mutant 18. Thus, the observed discrepancy is likely reflective of the error of the experiment, as suggested by Reviewer #2.

Reviewer #3 (Remarks to the Author):

The manuscript by Jones et al. used cryo-EM imaging/reconstruction and a surrogate system to determine the structure of *Klebsiella pneumoniae* capsulin and the binding site of its cargo-loading peptide. Mutagenesis studies were performed to pinpoint the residues in the peptide critical for binding. This is a solid work that details the structural basis of this particular capsulin/cargo-loading peptide pair, which is likely shared across a wide range of capsulin/cargo systems in many bacteria. However, similar structure and binding site information have already been reported in the original X-ray structure of *Thermotoga maritima* capsulin and its cargo-loading peptide binding site (Sutter ... Ban (2008). Structural basis of enzyme encapsulation into a bacterial nanocompartment. *Nature Structural & Molecular Biology*, 15(9), 939–947). While this manuscript has cited this paper, it does not include any comparison and it does not discuss the common findings and new knowledge generated by the new study.

Author reply: We have expanded our discussion of the paper highlighted by Reviewer #3, which we of course cited already, and have included more detailed comparisons with other TPs characterized so far.

Page 20, line 459 (discussion): “Compared with previously structurally characterized TP-shell interactions (*Thermotoga maritima*, Flp-TP: GGDLGIRK; *Haliangium ochraceum*, Flp-TP: GSLGIGSLR; *Myxococcus xanthus*, Flp-Tps: SHPLTVGSLRR, PEKRLTVGSLRR; *Quasibacillus thermotolerans*, IMEF-TP: TVGSLIQ)^{17,18,24,30} – all from ferroxidase cargos – the TP-shell interaction in our DyP system is mediated by three (Leu340, Ile342, and Leu345) instead of two hydrophobic residues. These three residues are accommodated by three distinct hydrophobic pockets located in the TP binding site with a defined register as highlighted by our mutational studies. Our high-resolution structure further allowed us to define H-bonding and ionic interactions, not previously described, also contributing to TP binding. Specifically, the C-terminal

lysine (Lys346) forms a salt bridge with the conserved TP-binding site residue Asp38 while the binding site residues Arg34 and Asp229 form hydrogen bonds with the TP backbone. A unique feature of the *T. maritima* TP – an intra-TP salt bridge between its aspartate and arginine residues, is not observed in our or any other TP structure. However, our DyP TP exhibits an intramolecular hydrogen bond between residues Asn341 and Ser339 which, in analogy to the salt bridge observed in the *T. maritima* system, appears to stabilize the observed TP conformation.”

A few minor points:

Fig. 1d: What is the unlabeled dashed curve?

Author reply: We thank Reviewer #3 for highlighting this oversight. The dashed line represents the absorbance at 320 nm. We have included the correct label into the figure panel and have updated the figure caption.

Have the authors tried to estimate the number of DyP hexamers in each Enc?

Author reply: Yes, based on gel densitometry, two DyP hexamers are encapsulated inside the encapsulin shell on average. It is certainly expected though that this number will vary from one to three DyP hexamers. Three hexamers represent the maximum number of DyP complexes per shell simply based on DyP complex size/volume and the available space inside the shell. We have added a sentence highlighting this fact to the results section.

Page 7, line 163: “Based on gel densitometry analysis, on average 12 copies of KpDyP are present per 60mer KpEnc shell, suggestive of encapsulation of two KpDyP hexamers per KpEnc.”

Fig. 2d: Why has DyP in Enc lost ~25% of heme while free DyP is nearly 100% loaded with heme? This seems in conflict with the statement “one additional function of encapsulation may be the stabilization of the most catalytically active oligomerization state of DyP” (p. 21, lines 474-6).

Author reply: This is a result of heterologous expression and the relative kinetics of shell self-assembly, cargo loading, and cofactor binding to the cargo. In general, encapsulin shell assembly and cargo loading occur concomitantly. The extent of cargo loading highly depends on the absolute and relative expression levels of both shell and cargo. If a cargo protein has to be loaded with a cofactor (for example heme, as in DyP), this adds another layer to the assembly of functional cargo-loaded shells. Cofactor loading of enzymes during heterologous overexpression is oftentimes not ideal. We often find that cofactor loading lags behind cargo-loading resulting in less than stoichiometric cofactor incorporation into encapsulated cargo proteins.

The cryoEM reconstruction of free DyP is only solved at 3.9 Å, a low-resolution structure for a hexamer of ~244 kDa protein complex with D3 symmetry. Sub-3 Å structure is expected with current cryo-EM technology.

Author reply: We have collected a new free KpDyP dataset. Instead of a Glacios microscope, a Krios instrument was used. The number of movies and resulting number of usable particles was increased/improved in this novel dataset resulting in a final resolution for the DyP hexamer of 2.39 Å. We have updated the relevant Supplementary Information figure(s) and table accordingly and have attached a new PDB validation report. Essentially no changes needed to be made to the atomic model.

Fig. 3f: How reliable are the 2D class averages shown in Fig. 3f? Have the authors tried to solve the 3D structure of the loaded DyP using the shell-subtracted images? A low-resolution 3D reconstruction should be possible if the 2D class averages are meaningful.

Author reply: The 2D classes are exclusively used to show that substantial amounts of KpDyP are loaded inside KpEnc, as outlined in the manuscript. They could not be used to generate meaningful ab-initio volumes for further downstream refinements.

Reviewer #4 (Remarks to the Author):

The manuscript "Structural basis for peroxidase encapsulation in a protein nanocompartment" by Jones et al. describes the structural and molecular basis for the encapsulation of the dye-decolorizing peroxidase (DyP) hexameric complex inside the encapsulin nanocompartment from the Gram-negative pathogen *Klebsiella pneumoniae* (named KpEnc). Using cryo-EM, the authors determined the structure of the DyP-loaded KpEnc and the free DyP hexamer (expressed in *E. coli*) at 2.5 and 3.9 Å resolution, respectively. DyP encapsulation is mediated by a targeting motif (named TP), located at its C-terminus, which recognizes a specific Enc binding pocket. However, this interaction is unresolved in the native, icosahedral DyP-loaded KpEnc due to the low TP occupancy. This drawback is smartly solved using a chimeric, non-native cargo that includes the DyP TP. This small-size cargo is encapsulated with a high copy number, and the TP-pocket interaction is described in the KpEnc system at 2.4 Å resolution. An exhaustive TP mutagenesis analysis establishes with precision the role of each residue and/or group of similar residues toward cargo encapsulation.

As would be anticipated from this group, the work presented is of excellent quality and advances the structural description of encapsulins in an incremental manner. I recommend publication, but suggest that the following very minor issues be (optionally) addressed:

1. To highlight this work, I would change the title of the manuscript to "Structural basis for peroxidase encapsulation inside the encapsulin from the Gram-negative pathogen *Klebsiella pneumoniae*"

Author reply: We thank reviewer #4 for this suggestion and have changed the title of the study accordingly.

2. The descriptions of T1, T3 or T4 (l. 45) are not formal to refer to triangulation numbers; I would specify the first citation as a T=1 shell, indicating the adopted nomenclature of T1 and so on.

Author reply: We thank reviewer #4 for this suggestion and have made the requested change to the introduction section.

Page 2, line 44: "Encapsulin shell proteins self-assemble into 18-42 nm protein shells consisting of 60, 180, or 240 identical subunits and exhibit icosahedral symmetry with triangulation numbers of T=1 (T1), T=3 (T3), or T=4 (T4)²³."

3. Following my general description above, and as described also by the authors in the abstract and at the end of the introduction, in my opinion, it would be more logic to show the DyP-loaded KpEnc and the free DyP hexamer structures in this order.

Author reply: We thank reviewer #4 for this suggestion. We agree and have rearranged the sections in question.

REVIEWERS' COMMENTS

Reviewer #2 (Remarks to the Author):

All of my comments have been addressed satisfactorily, and I appreciate the authors' time and efforts to do so.

Reviewer #3 (Remarks to the Author):

The manuscript has been revised to address previous concerns satisfactorily and I support its publication.

Reviewer #4 (Remarks to the Author):

In this revision, the authors have addressed properly the raised questions by all referees, not only my minor comments. The manuscript should be published in Nat. Comm.

Point-by-point response to reviewer comments:

Author replies shown in blue

Reviewer #2 (Remarks to the Author):

All of my comments have been addressed satisfactorily, and I appreciate the authors' time and efforts to do so.

Author reply: We thank Reviewer #2.

Reviewer #3 (Remarks to the Author):

The manuscript has been revised to address previous concerns satisfactorily and I support its publication.

Author reply: We thank Reviewer #3.

Reviewer #4 (Remarks to the Author):

In this revision, the authors have addressed properly the raised questions by all referees, not only my minor comments. The manuscript should be published in Nat. Comm.

Author reply: We thank Reviewer #4.